Assessment tools and incidence of hospital-associated disability in older adults: a rapid systematic review

Giacomino Katia katia.giacomino@hevs.ch 1 2
Hilfiker Roger 2
Beckwée David 1
Taeymans Jan 3 4
Sattelmayer Karl Martin 2
1 Department of Physiotherapy, Human Physiology and Anatomy, Faculty of Physical Education and Physiotherapy, Rehabilitation Research (RERE) Research Group, Vrije Universiteit Brussel , Brussels , Belgium
2 School of Health Sciences, HES-SO Valais-Wallis , Leukerbad , Valais-Wallis , Switzerland
3 Division of Physiotherapy, Department of Health Professions, University of Applied Sciences Bern , Bern , Switzerland
4 Faculty of Physical Education and Physiotherapy, Vrije Universiteit Brussel , Brussels , Belgium
Pawar Ajinkya
Electronic publication date: 2023 Oct 19
Publication date: 2023
Volume: 11
Electronic Location ID: e16036
Received 2023 Mar 29; Accepted 2023 Aug 14
Copyright: ©2023 Giacomino et al.
Copyright year: 2023
Copyright holder: Giacomino et al.
License: This is an open access article distributed under the terms of the Creative Commons Attribution License, which permits unrestricted use, distribution, reproduction and adaptation in any medium and for any purpose provided that it is properly attributed. For attribution, the original author(s), title, publication source (PeerJ) and either DOI or URL of the article must be cited.
License URL: https://creativecommons.org/licenses/by/4.0/

Keywords: Functional decline, Activities of daily living, Older patients, Hospitalization

Funding: The authors received no funding for this work.

==============================
Background

During hospitalization older adults have a high risk of developing functional impairments unrelated to the reasons for their admission. This is termed hospital-associated disability (HAD). This systematic review aimed to assess the incidence of HAD in older adults admitted to acute care with two outcomes: firstly in at least one activity of daily living from a set of functional tasks (e.g., Katz Index) and secondly the incidence of functional decline in an individual functional task (e.g., bathing), and to identify any tools or functional tasks used to assess activities of daily living (ADL) in hospitalized older patients.

Methods

A rapid systematic review was performed according to the recommendations of the Cochrane Rapid Reviews Methods Group and reported the data according the PRISMA statement. A literature search was performed in Medline (via Ovid), EMBASE, and Cochrane Central Register of Controlled Trials databases on 26 August 2021. Inclusion criteria: older adults (≥65 years), assessment of individual items of activities of daily living at baseline and discharge. Exclusion criterion: studies investigating a specific condition that could affect functional decline and studies that primarily examined a population with cognitive impairment. The protocol was registered on OSF registries (https://osf.io/9jez4/) identifier: DOI 10.17605/OSF.IO/9JEZ4.

Results

Ten studies were included in the final review. Incidence of HAD (overall score) was 37% (95% CI 0.30–0.43). Insufficient data prevented meta-analysis of the individual items. One study provided sufficient data to calculate incidence, with the following values for patients’ self-reported dependencies: 32% for bathing, 27% for dressing, 27% for toileting, 30% for eating and 27% for transferring. The proxy reported the following values for patients’ dependencies: 70% for bathing, 66% for dressing, 70% for toileting, 61% for eating and 59% for transferring. The review identified four assessment tools, two sets of tasks, and individual items assessing activities of daily living in such patients.

Conclusions

Incidence of hospital-associated disability in older patients might be overestimated, due to the combination of disease-related disability and hospital-associated disability. The tools used to assess these patients presented some limitations. These results should be interpreted with caution as only one study reported adequate information to assess the HAD incidence. At the item level, the latter was higher when disability was reported by the proxies than when it was reported by patients. This review highlights the lack of systematic reporting of data used to calculate HAD incidence. The methodological quality and the risk of bias in the included studies raised some concerns.

Introduction

Functional decline in older adults during hospitalization increases the risk of a longer hospital stay (Palmer et al., 1994), a nursing home placement (Fortinsky et al., 1999), and increased mortality (Brown, Friedkin & Inouye, 2004). The main goal in older adult care is therefore to maintain function (Izquierdo, Duque & Morley, 2021) and the ability to perform activities of daily living (ADL) (Covinsky et al., 1998; Palmer et al., 1994). During hospitalization older adults are at risk of developing functional decline unrelated to the condition for which they were admitted (Creditor, 1993). The loss of independence in at least one activity of daily living is referred to as hospital-associated disability (HAD) (Covinsky, Pierluissi & Johnston, 2011). Furthermore, HAD refers to disability acquired during hospitalization or the worsening of a pre-existing disability due to hospitalization (Covinsky, Pierluissi & Johnston, 2011).

Previous studies have highlighted methodological issues in assessing functional decline (Buurman et al., 2011; Loyd et al., 2020). However, there is currently no consensus on which tool should be used to assess functional decline in these patients, which ADL tasks should be included, how the assessment should be performed (self-reported or performance-based), and what time-frame should be considered (Buurman et al., 2011). Covinsky, Pierluissi & Johnston (2011) recommend asking patients on admission about their ADL functioning before the onset of acute illness.

A previous study highlighted the magnitude of the HAD problem and reported that the overall prevalence of HAD among older adults admitted to an acute care hospital is 30% (Loyd et al., 2020). To the best of our knowledge, the incidence of HAD has not been studied in a systematic review. The aims of this study was to perform a rapid systematic review to: (i) assess the incidence of HAD in older adults admitted to acute care with two outcomes: firstly in at least one activity of daily living from a set of functional tasks (e.g., Katz Index) and secondly the incidence of functional decline in an individual functional task (e.g., bathing), and (ii) identify any tools or functional tasks used to assess ADL in hospitalized older patients.

Materials & Methods

Registration

The protocol of this rapid systematic review registered at OSF registries with the following identifier: DOI 10.17605/OSF.IO/9JEZ4. Portions of this text were previously published as part of a preprint (https://www.medrxiv.org/content/10.1101/2022.09.22.22279726v1.full).

Study design

A rapid systematic review was performed according to the recommendations of the Cochrane Rapid Reviews Methods Group (Garritty et al., 2021). Reporting was conducted in accordance with the PRISMA statement (Page et al., 2021).

Search strategy and selection criteria

A literature search was performed in Medline (via Ovid), EMBASE, and Cochrane Central Register of Controlled Trials (CENTRAL) databases on 26th August 2021. All authors were involved in the literature search. The search strategy comprised five search terms related to: (i) study setting (i.e., hospital), (ii) observed disability in ADL, (iii) incidence and prevalence, sensitivity to change and responsiveness, (iv) population identification (i.e., older adults), and (v) articles that cover the aspect of disability acquired in hospitals. The search terms, combined with the Boolean operator “AND”, were applied to titles and abstracts, and MeSH terms were added when available and relevant. The full search strategy is shown in the additional material (see Table S1). The review included prospective and retrospective cohort studies. The control group of randomized controlled trials (RCTs) was eligible when performing a usual or a sham intervention. Inclusion criteria were: studies investigating a general older population (≥65 years) who were admitted to hospital for an acute disorder; studies had to assess the individual items of the ADL measurement tool before hospitalization (retrospectively or prospectively) and at the end of hospitalization or after hospital discharge. Exclusion criteria were: studies investigating a specific condition that could have an effect on functional decline (e.g., stroke, brain injury, heart failure, COVID-19, and acute respiratory failure); and studies that primarily examined a population with cognitive impairment.

Table 1 Study characteristics.

Author	Country	Sample size, n	Age, years, mean (SD) or (range)	Type of ward	Proportion of women, %  	% Living alone	% Living in nursing home	APACHE II score mean (SD) or % of patients per category	Charlson Comorbidity Index mean (SD) or % of patients per category	Mean number of days of hospitalization (SD) or (range)	Mental status mean (SD), tool) or % of patients with cognitive impairment	
Covinsky et al. (2000)	USA	2877	80.5	Medicine	64%	52%	5.1%	NR	NR	NR	1.4 (1.3), SPMSQ	
Covinsky et al. (2003)	USA	2293	79.5	Medicine	63.6%	35.2%	4.9%	0–2: 32.3%,
3–5 37.1%,
>6: 30.8%	0: 19.8%,
1–2: 47.0%,
3–4: 22.1%,
>5: 11.1%	6.3	NR	
Dharmarajan et al. (2020)	USA	515	82.7 (5.6)	Hospital	65.6%	46.6%	0%	NR	NR	NR	17.9%b, MMSE	
Hirsch et al. (1990)	USA	71	84 (75–95)	Medicine	59%	7.0%	5.6%	NR	NR	10.1 (2-49)	30.8%clast item of CNA	
Inouye et al. (1993)	USA	188	78.4 (5.8)	Medicine	59%	45%	4%	13.9 (3.6)	8.5 (2.8)a	7d (2 - 51)	23.5 (5.4), MMSE	
Martinez-Velilla et al. (2021)	Spain	149	87.1 (5.2)	Hospital	59%	NR	NR	NR	NR	8d	NR	
Mudge, O’Rourke & Denaro (2010)	Australia	615	80.4 (7.5)	Medicine	59%	NR	11%	NR	NR	7 (5–13)	10%, history of dementia	
Park et al. (2021)	Republic of Korea	RB: 45
PF: 36
MMF: 37
SF: 58	RB: 77 (73–82)
PF: 80 (74–84)
MMF: 81 (77–86)
SF: 81 (75–84)	Hospital	RB: 31.1%
PF: 38.9%
MMF: 46.0%
SF: 39.7%	NR	RB: 0%
PF: 2.8%
MMF: 8.1%
SF: 53.5%	NR	NR	NR	RB: 82.2%
PF: 58.3%
MMF: 70.3%
SF: 82.8%	
Sager et al. (1996)	USA	1279	79 (6.3)	Hospital	62%	37%	0%	NR	NR	8.6 (6.8)	17 (4.0), MMSE	
Zelada, Salinas & Baztan (2009)	Peru	GU: 68
UU: 75	GU: 79.6 (6.8)
UU: 76.1 (7.2)	Geriatric, 
usual unit	GU: 61.8%
UU: 56%	NR	NR	GU: 9 (2.87)
UU: 8.4 (3.11)	GU: 3.6 (1.98)
UU: 3.1 (1.6)	GU: 7.5 (4.3)
UU: 9.92 (7.74) 	GU: 22.4 (6.35)
UU: 2.7 (1.91), MMSE	
Notes.

a modified version.

b percentage of persons with a MMSE score <24.

c MMSE <19 points.

CNA nine-item care needs assessment

d median

GU geriatric unit

MMF mild-to-moderate frailty

MMS Mini-Mental State Examination

NR not reported

PF pre-frail

RB robust

SF severe frailty

SPMSQ Short Portable Mental Status Questionnaire

UU usual unit

One reviewer (GK) independently screened all the records based on titles and abstracts (phase 1) and the full-text of the eligible studies (phase 2). A second reviewer (SKM) screened 20% of the same records. If Cohen’s kappa coefficient of agreement was >0.80 for both phases, only 20% of the studies were planned to be screened independently by two reviewers. Disagreements were resolved through discussion.

Data extraction

Study characteristics (authors, country, study sample size, population age, type of ward, proportion of women, proportion living alone, proportion living in a nursing home, Acute Physiology and Chronic Health Evaluation (APACHE II) score, Charlson Comorbidity Index, number of days of hospitalization, any scale of mental status) were extracted by one reviewer (GK), while incidence data were extracted by three reviewers (HR, SKM, GK) at the same time. Disagreements were resolved through discussion between all three reviewers. The following information was extracted: item type, response options, criteria for the response options, baseline, and discharge assessment (i.e., who performed it and how), baseline and discharge prevalence of ADL dependency, operationalization, and definition of HAD and incidence per item and for the overall score.

Methodological quality assessment

Methodological quality assessment was performed using the critical appraisal checklist for Studies Reporting Prevalence Data (Munn et al., 2015) from the Joanna Briggs Institute (JBI). The JBI checklist was completed independently by two reviewers (SKM, GK). Differences in rating were discussed and resolved. We considered the methodological quality and the risk of bias of a study as low if all questions were rated ‘yes’, some concern if one of the questions was rated ‘unclear’ and high if one of the questions was rated ‘no’.

Incidence calculation of hospital-associated disability

The incidence of HAD (total score) was pooled using the statistical software R (R Core Team, 2022) and its package meta (Schwarzer, 2007). A random effects model was applied based on an inverse variance model with a logit transformation.

Heterogeneity was assessed by the I2, which is the proportion of total variability due to between-study heterogeneity (Deeks, Higgins & Altman, updated February 2022), to estimate inter-study variability. Tau was estimated with the DerSimonian-Laird estimator (Schwarzer, 2007). Migliavaca et al. (2022) recommended avoiding the use of an arbitrary cut-off for the I2 statistics, as it may not be discriminative in incidence studies. Therefore, heterogeneity was explained by discussing the level of dependency at baseline and the method of assessment.

The data required to calculate the incidence of HAD are the number of patients who are dependent and independent at baseline and the evolution of these groups at discharge. If these values were given, the following formula was used to calculate the incidence of the individual ADL task or set of tasks (total score): HADincidence=ab

where a and b represents the number of newly dependent patients (requiring the help of someone else), and the number of patients at risk of developing or increasing dependency respectively. For example, in the study by Inouye et al. (1993), 51 patients were newly dependent at discharge according to the overall score and 188 patients had the potential to decline in one of the ADL tasks. HAD incidence=51188=0.27.

If these values were not reported, an unbiased estimate of the incidence of HAD in that study could not be calculated. In this case, an estimate of the incidence was made by subtracting the percentage of dependency at discharge minus the percentage of dependency at baseline.

Tools used for assessment of activities of daily living in hospitalized older patients

ADL assessment tools are made up of several items each evaluating individual tasks (e.g., bathing, dressing, transferring, etc.) which together form a tool. The Barthel or Katz Index is an example of a tool composed of several tasks. The literature also mentions task sets that are not defined by the original authors as an index or tool. Other authors propose a set of individual tasks without creating an index.

The ADL tools assess whether patients are dependent or independent. For example, the Barthel Index scores the individual tasks from 0 points (dependent) to independent with a maximum of 15 points for transferring, 10 points for climbing stairs and 10 points for using the toilet.

All ADL tools or sets of tasks used in the assessment of hospitalized older patients were reported narratively.

Results

A total of 2,519 records were identified (Medline (via Ovid) 740 records, EMBASE 1,557, CENTRAL 222). After removing 743 duplicates, titles and abstracts of 1,776 articles were screened and 1,431 were excluded. The full text of 345 studies was screened, and a final total of 10 studies were included for further analysis (Covinsky et al., 2000; Covinsky et al., 2003; Dharmarajan et al., 2020; Hirsch et al., 1990; Inouye et al., 1993; Martinez-Velilla et al., 2021; Mudge, O’Rourke & Denaro, 2010; Park et al., 2021; Sager et al., 1996; Zelada, Salinas & Baztan, 2009). The reasons for exclusion are shown in the study flow diagram (see Fig. 1).

Figure 1 PRISMA flow diagram.

Cohen’s kappa coefficient was >0.8 in both screening phases (i.e., titles/abstracts and full texts); therefore only 20% of studies were screened by two reviewers independently, as described above.

Table 1 shows the population characteristics of the included studies. Six studies were performed in the USA, one in Europe, and three in other countries. The sample size ranged from 36 to 2,877 participants and the age of study participants ranged from 77 to 87 years. The proportion of women in the study samples under investigation ranged from 31% to 66%. Included studies were conducted in the medicine ward (five studies), general hospital (four studies), and one in the geriatric unit and usual unit which refers to a conventional care unit.

Outcome results

The pooled incidence of HAD for the overall score included two studies reporting the Katz Index of ADLs 1963 (Covinsky et al., 2003; Inouye et al., 1993) and two others used the Katz Index of ADLs 1970 (Mudge, O’Rourke & Denaro, 2010; Sager et al., 1996) for a total of 4,020 patients. Figure 2 presents the pooled incidence of HAD (total score) of 37% (95% CI [0.30–0.43]). Heterogeneity was substantial at 90%.

Figure 2 Forest plot of the incidence of overall hospital-associated disability.

The calculated and estimated incidences of HAD are categorized per ADL task and set of tasks (see Table S2). Calculated and estimated incidences of HAD categorized per ADL task and set of tasks). Adequate data for the incidence calculation at the item level was reported only by one study (Covinsky et al., 2000). These authors separated the patient reported and the proxy reported dependency in ADL. The incidence of HAD for the individual items was respectively 32% and 70% for bathing, 27% and 66% for dressing, 27% and 70% for toileting, 30% and 61% for eating, 27% and 59% for transferring. For the other studies, it was not possible to calculate the incidence of HAD (either total or single-item level) due to insufficient data. One RCT could not be integrated into Table S2 because the authors reported only the mean change in ADL score, and the results of this study were described narratively (Martinez-Velilla et al., 2021). The control group worsened in all items of ADL, negative values indicating a decline in the Barthel mean change score. The three items with the highest mean change score from baseline to discharge were: transferring (−2.06 points; 95% CI −1.44 to −2.71), climbing stairs (−1.36 points; 95% CI −0.84 to −1.91), and toileting (−1.23 points; 95% CI −0.76 to −1.69).

Ten studies were included in the identification of the tools or functional tasks used to assess ADL in hospitalized older patients (Covinsky et al., 2000; Covinsky et al., 2003; Dharmarajan et al., 2020; Hirsch et al., 1990; Inouye et al., 1993; Martinez-Velilla et al., 2021; Mudge, O’Rourke & Denaro, 2010; Park et al., 2021; Sager et al., 1996; Zelada, Salinas & Baztan, 2009). The review identified four assessment tools, two sets of tasks, and individual items added to original tools. The identified tools were: two references of the Katz Index (1970, 1963), the Barthel Index (Mahoney & Barthel, 1965), the nine-item Care Needs Assessment tool (Hirsch et al., 1990) and 7 ADL tasks form the Frailty Index (Rockwood & Mitnitski, 2011). The first set of tasks is composed of 2 activities coming from the Nagi (1976) and the Rosow & Breslau scale (1966) which were integrated in the Frailty Index (Rockwood & Mitnitski, 2011). The second sets of tasks was proposed by Gill (2014) (bathing, dressing, transferring and walking across the room). Walking was assessed in one study (Mudge, O’Rourke & Denaro, 2010), while one other study (Inouye et al., 1993) assessed the task of grooming by referring to the Katz Index of ADLs. However, the item walking and grooming were not in the original version of the Katz Index, and were considered separately.

Methodological quality

Figure 3 presents a summary plot of the assessment of methodological quality using the JBI checklist. All the included studies and study sub-groups reported an adequate sample frame and recruitment procedure. Sixty percent of the included study sub-groups reported an inadequate study sample size. All studies and sub-groups described in detail the study subjects and setting. Adequate analysis with sufficient coverage of the identified sample was conducted in 20% of studies sub-groups. All study sub-groups were unclear regarding the methods used for identification of the condition. Sixty percent of study sub-groups were rated unclear for the reliability of the measurement of participant’s condition. The statistical analysis was inadequate in approximately 86% of sub-groups. The response rate of the subgroups was unclear or inadequate in 86%.

Figure 3 JBI-Checklist summary plot.

Overall, the Covinsky et al. (2000) study showed some concern for methodological quality. All the other studies showed a high risk of bias. The rating of the individual items for each study is available in the additional material (see File S3).

Discussion

The aims of this systematic review were to evaluate the overall incidence of HAD as well as the incidence of HAD at the items level and to identify the tools or sets of tasks used in a hospital setting to assess ADL in older people.

The pooled overall incidence of HAD (total score) was 37%. Inouye et al. (1993) and Covinsky et al. (2003), who used the version of the Katz Index published in 1963, showed a lower overall incidence of HAD between 27% and 35% compared to Mudge, O’Rourke & Denaro (2010) and Sager et al. (1996). Mudge, O’Rourke & Denaro (2010) and Sager et al. (1996) who used the version of the same instrument published in 1970, showed a higher incidence of between 40% and 43% overall incidence of HAD compared to Inouye et al. (1993) and Covinsky et al. (2003). This difference could be explained by the fact that patients in the studies by Mudge, O’Rourke & Denaro (2010) and Sager et al. (1996) were more dependent at baseline compared with the participants included in the studies by Inouye et al. (1993) and Covinsky et al. (2003). We were unable to investigate whether this difference was explained by comorbidities (e.g., with the APACHE II score or the Charlson Comorbidity Index) as the authors of the studies by Sager and Mudge did not report these values.

Loyd et al. (2020) found a prevalence of HAD of 30% in older adults hospitalized for acute care. The current review focused on the number of new cases of HAD, i.e., the incidence of HAD. This latter measure considers the number of new cases of decline over the population at risk of developing a new decline, whereas prevalence takes into account the number cases of decline over the overall population. To our knowledge, no other review has neither investigated the overall incidence of HAD nor the incidence at the item level in a population of older patients over 65 years hospitalized for acute care. In general, the incidence of HAD at the item level was higher when disability was reported by the proxies than when it was reported by patients. Similar results were found in previous research (Rubenstein et al., 1984).

Loyd et al. (2020) included 15 articles, of which we included three in this present rapid systematic review (Hirsch et al., 1990; Inouye et al., 1993; Sager et al., 1996). Although all twelve studies addressed the issue of HAD, they were excluded on the basis of our inclusion criteria for the following reasons: (i) nine studies did not report the incidence of individual tasks (Boyd et al., 2009; Brown et al., 2016; Brown, Friedkin & Inouye, 2004; Chodos et al., 2015; Fimognari et al., 2017; Gill et al., 2004; Landefeld et al., 1995; Sourdet et al., 2015; Zaslavsky, Zisberg & Shadmi, 2015); (ii) three studies reported incidence of individual task but assessed functional declines between admission to discharge and not from baseline to discharge (McVey et al., 1989; Murray et al., 1993; Palese et al., 2016). None of the twelve studies reported individual ADL incidence or baseline assessment results in the supplementary or online literature.

Regarding the tools or sets of tasks used in a hospital setting to assess ADL in older people, this review identified that the Katz Index of ADLs was the most reported tool in the included studies. The Barthel Index and the Katz Index of ADLs are the oldest tools used for assessing ADL (Hartigan, 2007). Six studies used the Katz Index of ADLs, and none of them assessed the continence item that was include in the original version (Covinsky et al., 2000; Covinsky et al., 2003; Inouye et al., 1993; Mudge, O’Rourke & Denaro, 2010; Sager et al., 1996; Zelada, Salinas & Baztan, 2009). In addition, the Katz Index of ADLs was modified by adding another item of walking in two studies (Mudge, O’Rourke & Denaro, 2010; Sager et al., 1996) and adding the item grooming in one studies (Inouye et al., 1993).

An overestimation of HAD might be present for two main reasons. First, differences in the execution of the ADL assessment might have resulted in overestimation of the HAD incidence. In two studies (Inouye et al., 1993; Mudge, O’Rourke & Denaro, 2010), the baseline value for ADL ability was asked retrospectively to patients at admission, while ADL ability at discharge was assessed by an experienced healthcare professional. Older patients tend to overestimate their ADL ability (Kempen et al., 1996; Sager et al., 1992), their ability to step over an obstacle (Sakurai et al., 2013) and their motor performance (Kawasaki & Tozawa, 2020). Kawasaki and Tozawa hypothesized that the patients’ overestimation of their physical or functional capacities might be explained by the absence of recognition of their decline in motor performance (Kawasaki & Tozawa, 2020). Observer-based assessments of ADL tasks by healthcare professionals were more accurate than patients’ self-reported ADL values (Applegate, Blass & Williams, 1990; Elam et al., 1991).

Furthermore, this review included studies investigating pathologies that should not have a long-lasting effect on functional disability. However, it seems that there might be a combination of HAD and disease-related disability. For the overall HAD, Covinsky et al. (2003) and Mudge, O’Rourke & Denaro (2010) were the only ones to present the disability trajectories of each patient group and their development over time. These authors reported the percentage of patients whose ADL values declined between baseline (two weeks before hospitalization) and hospital admission (which is related to the condition) and those who had not recovered by discharge.

We believe that this combination of HAD and disease-related disability may also have contributed to an overestimation of HAD. In general, the studies did not separately report the numbers of persons with a disease-related disability and those with HAD, but only reported a combined total. Hence, it is unclear whether the failure to recover is due to hospitalization.

Moreover, the current ADL assessment tools present measurement properties limitations. For example, the Barthel Index showed a floor (Dromerick, Edwards & Diringer, 2003) and ceiling effect (de Morton, Keating & Davidson, 2008; Dromerick, Edwards & Diringer, 2003; Nielsen et al., 2016). Two of the studies included in the current review reported a ceiling effect of the Katz Index of ADLs (Mudge, O’Rourke & Denaro, 2010; Sager et al., 1996).

A strength of the current study is that, to the best of our knowledge, this is the first review to investigate the incidence of HAD in a population of older patients over 65 years of age, hospitalized for acute care. To our knowledge, previous systematic reviews did not assess the incidence of HAD at the item level.

The study has a number of limitations. The screening process was performed independently by two reviewers for only 20% of the records. However, we believe that this has only limited negative influence, as the agreement between the reviewers was high (above the predefined kappa coefficient of agreement of 0.8) and this procedure is accepted and suggested in rapid systematic review (Garritty et al., 2021). Another limitation is that we could not conduct a meta-analysis for the individual tasks due to insufficient data. With our reported methods, the certainty in the presented estimated incidence of HAD at the item level is very low. The estimated incidence, calculated as the difference between the discharge and baseline prevalence, should be interpreted with caution, as this does not consider the change over time of independent and dependent patients from baseline to discharge. The prevalence of disability at discharge does not distinguish between: (i) those who remain dependent between baseline and discharge, (ii) the newly dependent, and (iii) those who became independent at discharge. Therefore, these estimated incidences cannot be considered true incidence. In addition, small sample size bias cannot be totally excluded in the current review.

Implications and further research

Further studies should investigate the reasons for the overestimation of HAD. As reported above, there is a need to develop a more sensitive tool that reflects the true functional status of older patients (over 65 years) before hospitalization for acute care. The systematic integration of proxies in the evaluation of functional status before hospitalization, in addition to the patient self-reported assessment, needs to be deepened.

This study highlighted the fact that there is a lack of systematic reporting of data in order to assess the incidence of HAD in older patients aged over 65 years. This means that future cohort or intervention studies will have to report in detail the trajectory of such patients between before hospitalization, on admission and on discharge. Indeed, when retrospectively assessing patients’ ADL at home, it is necessary to detail the number of people being dependent for each item of ADL. On admission, it is important to describe which of the independent and dependent patients have improved, worsened, or maintained their status prior to hospitalization. At discharge, it is essential to describe which of the independent and dependent patients on admission have improved, worsened, or maintained their status compared with admission. Only in this way will it be possible to determine the number of people with a new decline and to determine its incidence. This review also raises public awareness of the need for careful interpretation of study data.

HAD is a relevant problem, and a systematic appraisal of existing intervention studies addressing this problem is missing. Future research should consider interview methods to help patients better remember their abilities at home in order to reflect their true ability in ADL function.

Conclusions

Functional decline in older patients over 65 years of age, due to hospitalization for acute care, is an important problem, with an incidence of 37% based on the overall score of ADL assessment. This incidence might be overestimated, due to a combination of disease-related disability and HAD, while measurement tools may also present some limitations. Furthermore, it is not possible to draw a definitive conclusion on the incidence of HAD at the item level, as there is insufficient data reported to enable the results of individual tasks to be pooled. This review highlights the lack of systematic reporting of data used to calculate the incidence of HAD. It is important to report the trajectory of dependent and independent people at each timepoints (baseline, admission, and discharge) and describe whether they have improved, deteriorated or maintained their status compared to the previous timepoint. Further studies should investigate the overestimation of HAD and how to overcome this limitation.

Supplemental Information

Supplemental Information 1 Raw data

The research strategy of all databases, detailed table describing the calculated and estimated incidences of HAD categorized per ADL task and set of tasks, the flow diagram and the individual question rating JBI Checklist.

Click here for additional data file.

Supplemental Information 2 JBI: Checklist study ratings of individual studies

Click here for additional data file.

Supplemental Information 3 PRISMA checklist

Click here for additional data file.

Supplemental Information 4 The rationale for conducting the systematic review

Click here for additional data file.

Additional Information and Declarations

Competing Interests

Author Contributions

Data Availability

The authors declare there are no competing interests.

Katia Giacomino conceived and designed the experiments, performed the experiments, analyzed the data, prepared figures and/or tables, authored or reviewed drafts of the article, and approved the final draft.

Roger Hilfiker conceived and designed the experiments, performed the experiments, analyzed the data, prepared figures and/or tables, authored or reviewed drafts of the article, and approved the final draft.

David Beckwée conceived and designed the experiments, performed the experiments, authored or reviewed drafts of the article, and approved the final draft.

Jan Taeymans conceived and designed the experiments, performed the experiments, authored or reviewed drafts of the article, and approved the final draft.

Karl Martin Sattelmayer conceived and designed the experiments, performed the experiments, analyzed the data, prepared figures and/or tables, authored or reviewed drafts of the article, and approved the final draft.

The following information was supplied regarding data availability:

The raw data are available in the Supplemental File.

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
