# Peer review of "Assessment tools and incidence of hospital-associated disability in older adults: a rapid systematic review"

_PeerJ, doi:10.7717/peerj.16036_

## Round 0.1 · original submission · Major Revisions

Please address to the comments raised.

Reviewer 1 ·

Basic reporting

Thank you for asking me to review #83733 ‘Assessment tools and incidence of hospital-associated disability in older adults: a rapid systematic review’
Hospital-associated disability is an important topic of growing interest. There is a 2020 review by Loyd et al which reviews the same topic, although the methods are somewhat different.

Overall the review is well written. The introduction mentions key early studies and the Loyd review, but justifies this additional review based on extraction of new data related to tools used.
1. Risk of bias is reported in several places eg line 232, 248 rather than just in the dedicated section line 274
2. The description of the Barthel index is out of place line 245-247 and would be better in describing tools, which should perhaps be the first part of the aim given the title of the paper. I think the authors could separate out description of the tools used from the analysis of sensitivity to change
3. I do not really understand the distinction between tools and tasks
4. The methodological quality section could be briefer, for example the sentence in line 278 could simply be “37% of studies reported an adequate sample size” as the remaining 63% can be inferred

Experimental design

Experimental design: The rapid systematic review methods are appropriately reported, with inclusion and exclusion criteria, an example of the search terms (although these appear very complex), a flow diagram, methodological assessment and statistical methods reporting. However:
6. I am unclear why the reviewers used only a selection of items from the methodological assessment to assess ROB as it is intended to be used in entirety
7. I am unsure of the validity of the methods used to evaluate ‘sensitivity to change’ (?responsiveness) of individual items

Validity of the findings

Validity of findings:
8. I am surprised at the lack of overlap with the studies reported in the previous Loyd review which needs explaining; there are a number of studies I would have expected this review to include which are absent.
9. I am struggling to understand number of studies included in the main outcome; there are 11 studies but only 4 appear to be contributing to the estimate of HAD, even though in additional file 2 there are 6 studies using composite ADL measures which appear to report a HAD rate
10. I do not think it is appropriate to include Hansen as this cannot be considered a ‘general older people’ cohort; it was a selected subgroup who were selected because they were independent at admissions and dependent at hospital discharge and thus inflates the prevalence
11. I cannot understand why ‘transferring’ is treated differently from ‘transferring from a chair’

Additional comments

General comments:
12. Line 300-302 should be ‘we were unable to investigate whether this difference was explained by comorbidities’ rather than that it could not be explained, which suggests that it was investigated and found negative
13. There is a lot of data about tools included in the supplementary material but if the emphasis is on the tools then this should be included in the main paper
14. There are some errors of fact on checking some of the primary papers e.g. line 265 Mudge et al used ‘mobility’ not ‘walking’ while Sager et al did use ‘walking’ (across a room); it is not true that only Covinsky’s 2003 study reported trajectories (line 356) as Mudge et al used similar methods and specifically reported pre-hospital decline, in hospital decline, and in hospital recovery.
15. I am not really sure what the item ‘sensitivity’ analysis really adds for clinical use, nor what this paper adds to the previous Lloyd review which had more studies; while conceptually the authors tried to be more precise about disease vs hospital-related decline, and removing dependent patients from the denominator, in both the additional table 2 and the discussion it is clear this was not able to achieved (perhaps partly because authors were not contacted for this more detailed data)

·

Basic reporting

Clear and unambiguous
References sufficient
Aim of the study explicitely mentioned

Experimental design

Its a rapid systematic review
Research question addressed, well defined and relevant
Some suggestions provided in the PDF

Validity of the findings

Minor changes suggested in Statistical part

---

## Round 0.2 · accepted · Accept

The authors have amended all the comments successfully
Congratulations